# Tea-Waste-Mediated Magnetic Oxide Nanoparticles as a Potential Low-Cost Adsorbent for Phosphate ($PO_4^{3-}$) Anion Remediation

Khizar Hussain Shah [1], Misbah Fareed [2], Muhammad Waseem [1], Shabnam Shahida [3], Mohammad Rafe Hatshan [4], Sadaf Sarfraz [5], Aneeqa Batool [2], Muhammad Fahad [6], Tauqeer Ahmad [7], Noor S. Shah [8], Kyungeun Ha [9] and Changseok Han [9,10,*]

[1] Department of Chemistry, COMSATS University Islamabad (Islamabad Campus), Islamabad 45000, Pakistan; khizar_nce@yahoo.com (K.H.S.); waseem_atd@yahoo.com (M.W.)
[2] Department of Chemistry, COMSATS University Islamabad (Abbottabad Campus), Abbottabad 22060, Pakistan; misbah.fareed1234@gmail.com (M.F.); aneeqabatool196@gmail.com (A.B.)
[3] Department of Chemistry, Faculty of Applied and Basic Sciences, University of Poonch, Rawalakot 12350, Pakistan; shabnamshahida01@gmail.com
[4] Department of Chemistry, College of Science, King Saud University, P.O. Box 2455, Riyadh 11451, Saudi Arabia; mhatshan@ksu.edu.sa
[5] Department of Chemistry, Lahore Garrison University, Lahore 54000, Pakistan; sadaf_sarfraz@hotmail.com
[6] Department of Electrical and Computer Engineering, COMSATS University Islamabad (Abbottabad Campus), Abbottabad 22060, Pakistan; drmuhammadfahad@gmail.com
[7] Department of Chemistry, University of Mianwali, Mianwali 42200, Pakistan; tskp72@yahoo.com
[8] Department of Environmental Sciences, COMSATS University Islamabad (Vehari Campus), Vehari 61100, Pakistan; noorsamad@cuivehari.edu.pk
[9] Program in Environmental & Polymer Engineering, Graduate School, INHA University, Incheon 22212, Republic of Korea; kyungeunha0402@gmail.com
[10] Department of Environmental Engineering, INHA University, Incheon 22212, Republic of Korea
* Correspondence: hanck@inha.ac.kr

**Abstract:** In the current study, magnetic oxide nanoparticle-impregnated tea waste (TW-$Fe_3O_4$) is employed as an adsorbent to remove phosphate ions ($PO_4^{3-}$) from an aqueous solution. By utilizing a variety of analytical methods, the TW-$Fe_3O_4$ nano-adsorbent was characterized by FE-SEM, TEM, EDX, BET, FTIR and XRD. The FE-SEM of TW-$Fe_3O_4$ demonstrated the adsorbent's granular morphology with a variety of magnetic nanoparticle sizes and shapes. The XRD of TW-$Fe_3O_4$ showed two diffraction peaks at 2θ values 30.9° and 35.4°, which are in correspondence with the diffraction pattern of magnetite. The synthesis of a TW-$Fe_3O_4$ adsorbent with a greater surface area and porosity was demonstrated by BET analysis. Numerous adsorption factors like initial concentration of $PO_4^{3-}$ ion, pH of the medium, contact time, temperature and adsorbent dose were optimized for phosphate removal. The maximum removal of 92% was achieved by using the adsorbent dose of 1.2 g at 323 K (pH 5). Pseudo-second-order and intra-particle diffusion models were fitted to the sorption kinetic, whereas adsorption isotherm data were found well fitted to Freundlich and Dubinin–Radushkevich (D-R) models. The highest adsorption capacity of TW-$Fe_3O_4$ towards phosphate ions was 226.8 mg/g, which is significantly higher than other reported bio-adsorbents. According to thermodynamic data, phosphate adsorption at the solid–liquid interface was of an endothermic and spontaneous nature and characterized by enhanced inevitability.

**Keywords:** adsorption; batch studies; characterization; phosphate; potential adsorbent; TW-$Fe_3O_4$

## 1. Introduction

The contamination of water bodies by toxic cationic and anionic species, organic dyes, pharmaceutical products and agricultural and industrial wastes has become a major concern throughout the world [1,2]. Among these toxicants, phosphate ($PO_4^{3-}$) ions have

attained great attention in wastewater management due to their negative impact on human beings and environmental safety.

The primary forms of phosphate ($PO_4^{3-}$) found in the natural environment include organic and inorganic phosphates, oligophosphates and polyphosphates [3]. It supports the growth of most biological organisms and many biochemical processes in aquatic environments. However, it is considered to be very dangerous when its concentration exceeds the recommended limit (1 $\mu g \cdot L^{-1}$) in water according to the United States Environmental Protection Agency (USEPA) [4]. An excess amount of phosphates compared to the permissible limit in water can cause water eutrophication and increase the population of pathogens, resulting in damage to water quality and deterioration of aquatic life [5,6].

To date, a number of analytical procedures, including reverse osmosis [7] electrodialysis [8], crystallization [9], ion exchange process [10], chemical precipitation [11] and adsorption [12], have been developed for the efficient removal of phosphate ions ($PO_4^{3-}$) from aqueous solutions. Among all the above-prescribed techniques, adsorption has several advantages such as high removal efficiency at both low and high contaminant concentrations, convenient operation and designing, lack of sludge formation and no production of secondary pollutants, ease of regeneration and reuse of adsorbent and adsorbate [12]. However, selecting a suitable, green and economical potential adsorbent is challenging in adsorption technology. Therefore, adsorbents obtained from biological materials or agricultural bi-products and their surface-modified forms have received excellent research attention in recent decades.

Tea waste (TW) is one of the promising biomass adsorbents for effective water decontamination because of its many organic functional sites and porous network matrix structure [13–16]. Besides this, it also has a wide range of availability at a low cost, is eco-friendly in nature and serves as a good adsorption potential. Moreover, tea waste is widely consumed in our societies and can be easily collected from local restaurants, cafes, and houses for secondary use as a raw material for chemical modification. Through chemical modification of tea waste, the physiochemical properties and contaminant removal efficiency of chemically modified tea waste can be successfully improved as compared to unmodified or native forms [14,17,18]. Therefore, chemically modified tea waste adsorbents were applied to remove different organic and inorganic contaminants from aqueous solutions and quite satisfactory results were obtained [14,19–22]. However, many chemical modifications of tea waste carry a huge cost, and the recovery of adsorbents from aqueous solutions is significantly more challenging for recycling and reusability. The introduction of $Fe_3O_4$ nanoparticles onto the surface of tea waste adsorbent is the best method to overcome these problems [16].

Nanotechnology is an emerging field of science with a broad range of innovative applications in various fields like medicines, electronics, energy and environmental sensing and wastewater treatment [23–30]. The integration of nanotechnology with adsorption technology has introduced a new category of potential and novel adsorbents with desirable surface characteristics and unique functionalities [26,27]. A variety of potential and economic metal oxide nanoparticles were synthesized and targeted for decontamination of water and wastewater solutions [26,27]. Among all other metal oxide nanoparticles, $Fe_3O_4$ nanoparticles have offered unique advantages and specific features like easy and green synthesis, high surface area, high separation tendency under an applied magnetic field, easy functionalization and biocompatibility [31]. Also, regeneration of $Fe_3O_4$ nanoparticles is very convenient, which leads to protection from economic loss [32].

The application of $Fe_3O_4$ nanoparticles as nano-adsorbents individually and in composite form has received great attention in the field of adsorption technology [16,33,34]. Recently, many reported studies have shown that the impregnation of magnetite ($Fe_3O_4$) nanoparticles onto the surface of tea waste can enhance the surface area, porosity, mechanical stability, easy magnetic separation and adsorption properties of tea waste biosorbent [16,33,34]. In parallel, the reported studies indicate that impregnation of $Fe_3O_4$ nanoparticles also improves the homogenous dispersibility and stability of $Fe_3O_4$ nanopar-

ticles onto the surface of tea waste [13]. It is noticeable that individual use of $Fe_3O_4$ nanoparticles as an adsorbent can create adsorbent loss due to their nanosized structure. Also, the recovery of adsorbent by simple gravitational sedimentation is difficult after adsorption to avoid secondary pollution in water bodies [35,36]. Therefore, impregnation of $Fe_3O_4$ nanoparticles onto a supporting solid biomass is compulsory, which enables the rapid recovery of the spent adsorbent from decontaminated water under the influence of an applied magnetic field. In addition to this, studies indicate that a composite form of $Fe_3O_4$ nanoparticles also provides an effective increase in surface area by providing plenty of functional sites for adsorption [19,36]. The overall synthesis of magnetic-oxide-impregnated tea waste (TW-$Fe_3O_4$) nano-composite adsorbent is very easy, economical and eco-friendly due to the utilization of low-cost and non-toxic materials. Also, the recovery of TW-$Fe_3O_4$ adsorbent is easy and fast with simple magnetic operation after adsorption from aqueous solutions [37]. Hence, magnetically modified tea waste adsorbents were potentially targeted towards the efficient removal of metallic cations, such as Ni (II), Pb (II), As (III), Hg (II) and U(VI), from water solutions, and excellent results were obtained [33,34,37–41]. However, limited literature is reported for the removal of Chromate ($CrO_4^{2-}$), Arsenate ($AsO_4^{3-}$) and Fluoride ($F^-$) anion adsorption by magnetically modified tea waste adsorbent [16,22,42–45]. As far as the removal of phosphate anions is concerned, no reasonable data are available from using $Fe_3O_4$-nanoparticle-impregnated tea waste. The current research work, therefore, was conducted with the objective of exploring the potential of $Fe_3O_4$-nanoparticle-impregnated tea waste as a low-cost adsorbent in removing phosphate ($PO_4^{3-}$) anions from aqueous solutions. The adsorbent was characterized by Field Emission Scanning Electron Microscopy (FE-SEM), Transmission Electron Microscopy (TEM), Energy-Dispersive X-ray (EXD), $N_2$ physisorption (BET), Fourier Transform Infrared Spectroscopy (FTIR) and X-ray diffraction (XRD). For adsorption studies, various parameters like initial adsorbate concentration, pH, temperature, adsorbent dose and contact time were evaluated. The adsorption data were tested through isotherms and kinetic models. The reusability of the adsorbent in up to six consecutive adsorption/desorption cycles ensured the effectiveness of $Fe_3O_4$-nanoparticle-impregnated tea waste adsorbent. To the best of our knowledge, the adsorption of phosphate ($PO_4^{3-}$) anions from aqueous solutions by using a TW-$Fe_3O_4$ absorbent is the first of its kind. Likewise, this study can also be helpful due to the disposal of phosphate-loaded exhausted adsorption media in agricultural lands, which can be used ultimately as a phosphate-bio-fertilizer by enriching soil fertility.

## 2. Experimental Section

### 2.1. Materials and Methods

In this work, the chemicals and reagents used were of analytical grade. Sodium dihydrogen phosphate ($NaH_2PO_4$), Ferrous sulphate ($FeSO_4 \cdot 7H_2O$), Ferric chloride ($FeCl_3 \cdot 6H_2O$), potassium dihydrogen phosphate ($KH_2PO_4$), ammonium molybdate (($NH_4)_2MoO_4$), ascorbic acid ($C_6H_3O_6$), sodium hydroxide (NaOH), hydrochloric acid (HCl), Sodium chloride (NaCl) and Potassium chloride (KCl) were purchased from Daejung (Republic of Korea). The tea waste samples (discarded Lipton tea bags) were collected from the local restaurants of Islamabad, Pakistan.

### 2.2. Preparation of TW-$Fe_3O_4$ Adsorbent

The tea waste was first washed thoroughly with distilled water and then was boiled to eliminate color and water-soluble impurities, if any. Finally, it was washed thoroughly with triple-distilled water and dried in an oven at 105 °C for 12 h. After that, it was ground to fine powder and stored in plastic vials after passing through a sieve with a size of 100 μm. The $Fe_3O_4$-TW adsorbent was prepared by a chemical precipitation method as reported in the literature [16,19]. For this purpose, 5.2 g of $FeSO_4 \cdot 7H_2O$ was dissolved in 40 mL of distilled water and stirred continuously. Ferric chloride solution was prepared by dissolving 7.4 g $FeCl_3 \cdot 6H_2O$ in 40 mL of distilled water and stirred continuously till it dissolved in water completely. Both solutions were mixed with each other and stirred

for 30 min at 80 °C, to which 20 mL of 25% ammonia solution was added dropwise till the pH value was 10.5. The color of the solution changed to blackish or dark brownish. For impregnation of tea waste, 10 g of purified tea waste was dispersed in the combined solution and was stirred vigorously for 30 min at 80 °C in order to ensure the complete growth of magnetite nanoparticles on the surface of tea waste. The whole process was performed under an inert atmosphere. Magnetized tea waste was cooled and allowed to settle down at room temperature, and was washed with triple-distilled water and absolute ethanol to remove the unreacted chemicals completely. An external magnetic rod was used to separate the synthesized $Fe_3O_4$-TW adsorbent from the solution. Finally, the prepared adsorbent was oven-dried under a vacuum at a temperature of 80 °C for about 12 h and then homogenized using a mortar and pestle. The powder obtained was finally stored for adsorption experiments.

### 2.3. Characterization

Different characterization techniques were applied to analyze the TW-$Fe_3O_4$ to successfully elucidate the adsorption of phosphate ions on the adsorbent surface. Fourier Transform Infrared (FTIR) spectra of the as-synthesized adsorbent sample recorded in the wavenumber range 4000–400 cm$^{-1}$ by the SCHIMADZU 8202PC (Kyoto, Japan) model confirmed the modification of tea waste. The surface morphology of both TW and TW-$Fe_3O_4$ were studied by Scanning Electron Microscopy (SEM) using a JEOL JSM 5120 SEM (Tokyo, Japan) operated at 20 kV accelerating voltage. The sample was carbon-coated before being put into the SEM to avoid charging. To investigate the specific pore size, an ASAP 2400 surface area and porosimetry system (Micromeritics) were used. The surface areas and the pore sizes were calculated by using BET and BJH methods.

### 2.4. Determination of Point of Charge Ratio (PZC)

The zero surface of the TW-$Fe_3O_4$ adsorbent was determined by using the salt addition method [17]. For this purpose, 30 mL of 0.1 M NaCl solutions was added to 50 mL conical flasks and the initial pH (pH$_i$) of solutions were adjusted in the range of 2–12 by using 0.01 M solutions of HCl/NaOH. In these solutions with different pH values, 0.2 g of adsorbent was poured into each of the flasks and stirred for one hour with the help of a magnetic stirrer. After mixing, a change in the final pH value (pH$_f$) of solution was measured. The measurement of pH of solutions was performed by using a pH meter model 700 (EUTECH, Singapore). The PZC of the adsorbent was obtained from the intersection point by plotting $\Delta$pH (pH$_i$-pH$_f$) versus the initial pH of the solution.

### 2.5. Batch Adsorption Experiments

The adsorption of phosphate ions by TW-$Fe_3O_4$ was studied using the batch adsorption technique, in which the effect of different parameters on the adsorption process was investigated. These parameters included contact time, variation in pH of the solution, dosage of the adsorbent concentration of adsorbate, and temperature during the adsorption process. The adsorption data were collected by the average of triplicated experiments of phosphate adsorption. Percent removal of phosphate ions with TW-$Fe_3O_4$ was calculated as follows:

$$\%R = \frac{C_{i-}C_e}{C_i} \times 100 \tag{1}$$

where $C_i$ (mg/L) is the initial concentration and $C_e$ (mg/L) denotes the equilibrium concentration of phosphate ions. The amount of phosphate adsorbed per unit mass of TW-$Fe_3O_4$ ($q_e$, mg/g) was also calculated by the mass balance equation:

$$q_e = \frac{V(C_{i-}C_e)}{1000m} \tag{2}$$

where $V$ is the volume of the solution (mL) and $m$ is the mass of the TW-$Fe_3O_4$ adsorbent (g).

### 2.5.1. Effect of Contact Time

In order to examine the influence of contact time, 1.2 g of TW-Fe$_3$O$_4$ was poured into 100 mL of 5 mg/L phosphate solution and placed in a thermostat shaking water bath, DAIHAN WSB-30 (Wertheim, Germany). The solution was shaken at 300 rpm and 298 K. The different sample solutions were taken at various times between 5 and 90 min. The adsorbent was then removed from the solution, and the filtrate samples were examined for phosphate concentration by the below-mentioned spectrophotometric method.

### 2.5.2. Effect of Adsorbate Concentration

To determine the effect of initial phosphate concentration on the adsorptive removal of phosphate ions, 1.2 g of TW-Fe$_3$O$_4$ was introduced into 100 mL of different initial phosphate concentrations (1–20 mg/L). After adjustment of optimized pH for each concentration, the suspensions were shaken at 300 rpm and 298 K for 75 min. After the attainment of equilibrium, the suspensions were filtered and analyzed for phosphate determination.

### 2.5.3. Influence of pH Values

The effect of pH on adsorption process was investigated by maintaining the pH range 2–12 in different conical flasks, each containing 100 mL of 5 mg/L phosphate solution at 298 K. Then, 1.2 g of adsorbent was added to each flask and the suspensions were shaken for 75 min at 300 rpm. After equilibrium time, the filtrate was analyzed for phosphate adsorption.

### 2.5.4. Influence of Adsorbent Dosage and Temperature

To investigate the effect of adsorbent amount on the adsorption process, different amounts of TW-Fe$_3$O$_4$ (0.2–1.5 g) were taken in different conical flasks containing 100 mL of 5 mg/L phosphate solution in each flask. The solutions in conical flasks with different amounts of TW-Fe$_3$O$_4$ were shaken at 300 rpm at pH 5 for 75 min. The residue of phosphate was analyzed after filtration. In a similar practice, the influence of temperature on the adsorption process was carried out by taking 1.5 g of TW-Fe$_3$O$_4$ in 100 mL of 5 mg/L phosphate solution at pH 5. The temperature of the suspension was maintained in the range of 298–333 K. The shaking rate and time were selected as above. Finally, the suspension was filtered and analyzed for phosphate ions.

### 2.5.5. Regeneration of TW-Fe$_3$O$_4$ Adsorbent

The reusability test of the adsorbent was performed by introducing 1.2 g phosphate-adsorbed TW-Fe$_3$O$_4$ into 100 mL of NaOH (1 M) and the suspension was shaken at 300 rpm and 298 K for 75 min. The concentration of phosphate for each adsorption–desorption cycle was measured to determine the removal proficiency of the adsorbent after six consecutive recycling instances.

### 2.5.6. Phosphate Determination

After each batch adsorption experiment, the determination of phosphate ions in aqueous phase was performed using the well-known ascorbic acid method with a UV–Vis spectrophotometer SPECORD PLUS 200 (Analytik Jena, Jena, Germany) by selecting the wavelength of almost 850 nm [35].

### 2.5.7. Analytical Figures of Merit

The calibration curve for determination of PO$_4^{3-}$ was found to be linear in the range of 0.031–12 mg/L, with a correlation coefficient ($R^2$) of 0.9951. Limit of detection (LOD), calculated as the concentration equivalent to three times the standard deviation of 10 blank measurements, was 0.009 mg/L. Similarly, limit of quantification (LOQ) was calculated as the concentration equivalent to ten times the standard deviation of 10 blank measurements, and was found to be 0.031 mg/L. Besides conventional parameters to estimate the sensitivity of the proposed procedure, method detection and quantification limits were also calculated.

The method detection limit (MDL) is the minimum amount of $PO_4^{3-}$ ion determined in an aqueous solution with 99% confidence, as the concentration was greater than 0. It was found by examining spiked aqueous samples at a level yielding $3 \times SD/n$; calculated as $(t_{(99\%, n=10)} \times SD)$. On the other hand, the method quantitation limit (MQL) was calculated as $3.34 \times MDL$. The MDL and MQL observed were 0.005 and 0.016 mg/L, respectively.

## 3. Results

### 3.1. Characterization Results

#### 3.1.1. FTIR

The FTIR spectra of TW and that of modified TW-$Fe_3O_4$ are shown in Figure 1a. The FTIR spectrum analysis for parent TW indicated a broad band at 3276 cm$^{-1}$ representing bonded -OH groups. The bands observed at 2919 and 2840 cm$^{-1}$ indicated the existence of an aliphatic C–H group. The shoulder band at wavenumber 1720 cm$^{-1}$ was observed due to the presence of C=O stretching vibration. The wave trough at 1622 cm$^{-1}$ indicated the C=O stretching mode conjugated with the $NH_2$, while the peak observed at 1031 cm$^{-1}$ was considered to be due to the S=O functional group. From the FTIR spectra, it was observed that the modification of TW with iron oxide nanoparticles caused the relevant bands to shift from 3276 and 1622 cm$^{-1}$ to 3336 and 1614 cm$^{-1}$, respectively. The modified tea waste showed a band at 545 cm$^{-1}$ due to the bending vibration of Fe-O which confirmed the presence of the Fe-O functional group in impregnated tea waste [13,34].

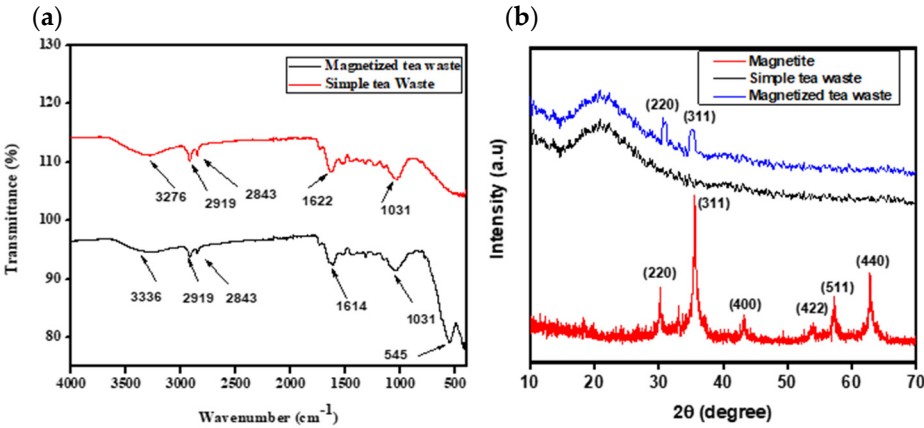

**Figure 1.** (**a**) FTIR and (**b**) XRD spectra of TW and TW-$Fe_3O_4$.

#### 3.1.2. XRD

To study the crystalline structure, XRD patterns of magnetite, tea waste and magnetite-loaded tea waste were recorded in a diffractometer model X'Pert PRO (PHILIPS, Amsterdam, The Netherlands). The scanning rate was 8°/min in the 2θ range 10° to 70° (Figure 1b). In magnetite, the diffraction peaks appeared at the 2θ values 30.9°, 35.4°, 43.3°, 53.9°, 57.2° and 62.9°, which were indexed with planes (220), (311), (400), (422), (511) and (440), respectively. The diffraction pattern confirms that the magnetite nanoparticles were well crystallized. All the diffraction peaks were matched well with the reference card (JCPDS 19-0629). The crystallite size was calculated by using Sherer's equation and it was about 10.6 nm. The small size of crystallites can be linked to the broadening of the diffraction peaks. A similar diffraction pattern was reported by Salem et al. while working on magnetite nanoparticles synthesized from ferrous ions and pistachio leaf extract [46,47]. In the case of tea waste samples, no diffraction peak was detected, suggesting that the surface was predominantly amorphous. The diffraction pattern of TW-$Fe_3O_4$ clearly showed only two diffraction peaks at the 2θ values 30.9° and 35.4°, which are indexed with the hkl planes (220) and (311), corresponding with the diffraction pattern of magnetite [34]. In TW-$Fe_3O_4$, only two diffraction peaks were observed, which shows that the growth of the magnetite nanoparticles predominantly took place in the 220 and 311 planes. These

reflections are well corrected with a spinel structure of magnetite with the JCPDS reference card 65-3107 [47].

### 3.1.3. SEM/EDX, FESEM and TEM

The SEM, FESEM and TEM images of pristine and magnetized tea waste are shown in Figure 2.

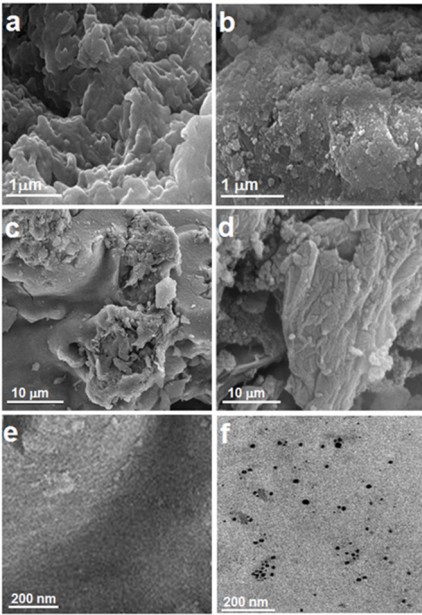

**Figure 2.** (**a**,**b**) SEM; (**c**,**d**) FESEM; and (**e**,**f**) TEM for TW and TW-$Fe_3O_4$.

Tea waste presented a stem structure, as its main components included cellulose and hemicellulose. The SEM images of parent tea waste showed that it had a heterogenous, rough and porous surface (Figure 2a,b). Moreover, the modified tea waste had more fibers and a rough surface suitable for adsorbing phosphate ions [13]. The surface morphology of samples was also observed by FESEM (Figure 2c,d) at 1000× resolution in a model ZEISS Supra 35VP. The images show that the surfaces of both TW and TW-$Fe_3O_4$ were non-uniform and porous in nature. However, the impregnation of magnetite nanoparticles was still invisible. To further explore the in-depth surface modification, TEM images of TW and TW-$Fe_3O_4$ were acquired and recorded with a model FEI Tecnai 12. To acquire the thin section, the samples were prepared by ultramicrotomy. The interaction of fine magnetite nanoparticles with tea waste can be shown clearly from the TEM images (Figure 2e,f). These results supplement the BET analysis where the surface area of TW-$Fe_3O_4$ was increased, which may be due to small-sized magnetite nanoparticles anchored on the surface. EDX analysis represents the elemental composition of both TW and TW-$Fe_3O_4$. The analysis confirmed the presence of 55.5% C, 15.8% O, 17.2% Si, 7.1% Ca, 3.1% Na and 1.3% Mg in the TW sample. After impregnation of $Fe_3O_4$ on tea waste, the elemental composition was 7.8% Fe, 53.4% C, 22.2% O, 11.3% Si, 2.1% Ca, 1.7% Na and 1.5% Mg. The elemental analysis showed the successful impregnation of $Fe_3O_4$ onto the surface of tea waste.

### 3.1.4. BET

To determine the surface area, specific pore size and average pore diameter of TW and $Fe_3O_4$-TW, both Brunauer–Emmett–Teller (BET) and Barrett–Joyner–Halenda (BJH) analyses were performed, and the data are shown in Table 1. It was observed that the BET and BJH surface areas of the TW increased two-fold from 20.17 to 45.13 $m^2$/g and 11.58 to 22.85 $m^2$/g, respectively, after impregnation by $Fe_3O_4$ nanoparticles. Further, an increase in the pore volume with decrease in the pore radius was observed, showing that more nitrogen molecules were absorbed, resulting an increase in the surface area. The increase in

surface area afterwards may be due to the enhanced growth of $Fe_3O_4$ on the surface as well as its impregnation within the tea waste [15,48].

**Table 1.** The BET surface area, pore radius and pore volume of TW and TW-$Fe_3O_4$.

| Material | Surface Area (m²/g) | | Pore Radius (A°) | Pore Volume (cc/g) |
| --- | --- | --- | --- | --- |
| | **BET** | **BJH** | **BJH** | **BJH** |
| TW | 20.17 | 11.58 | 13.01 | 0.010 |
| TW-$Fe_3O_4$ | 45.13 | 22.85 | 9.27 | 0.020 |

### 3.2. Adsorption Studies

The efficacy of TW-$Fe_3O_4$ was investigated through phosphate ion uptake under different adsorption parameters such as PZC, pH, adsorbent dosage, temperature, adsorbate concentration and contact time.

### 3.2.1. PZC

The point of zero charge (PZC) value of an adsorbent is the characteristic of an adsorbent where the net surface charge of the adsorbent is equal to zero. The PZC of TW-$Fe_3O_4$ adsorbent was determined to be pH 6 and the results are shown in Figure S1. When the pH of the solution was below PZC, the positive surface charge of adsorbent preferred anions for adsorption. Conversely, when the pH of the analyte solution was above $pH_{pzc}$, the negative surface charge of adsorbent preferred cations for adsorption.

### 3.2.2. Influence of pH

The pH of the solution is one of the most critical factors which has a significant impact on the adsorption process. The results are shown in Figure 3a, indicating that the adsorption strongly depends on the pH of the phosphate solution. It can be predicted that the removal efficiency of phosphate onto the TW-$Fe_3O_4$ adsorbent was enhanced from 62.3 to 91.8% when solution pH was increased from 2 to 5, while the removal efficiency was reduced from 91.8 to 54.7% when solution pH was increased from 5 to 9. The maximum removal efficiency (91.3%) of phosphate ions onto TW-$Fe_3O_4$ adsorbent was observed at pH 5.

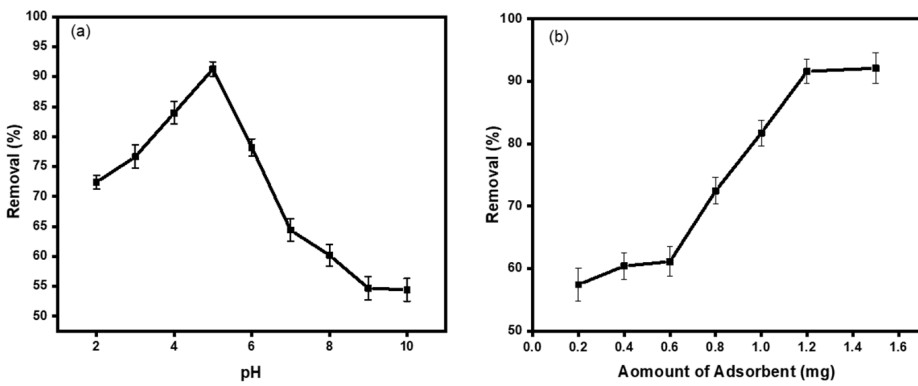

**Figure 3.** Effect of pH and adsorbent dosage on the percentage removal of $PO_4^{3-}$ by TW-$Fe_3O_4$.

The influence of pH on the adsorption of phosphate ions can possibly be explained based on the surface charge of the adsorbent and different phosphate species present in water at different pH values of the solution. As discussed earlier, the $pH_{pzc}$ of TW-$Fe_3O_4$ adsorbent was determined to be pH 6.1 before the phosphate adsorption. Therefore, a phosphate can exist in different anionic species in the aqueous phase, such as $H_2PO_4^-$, $HPO_4^{2-}$ and $PO_4^{3-}$, depending upon the pH of the solution. In an acidic solution, the phosphate exits in monovalent ($H_2PO_4^-$) and divalent $HPO_4^{2-}$ forms. However, in an alkaline solution, phosphate mainly exists in trivalent ($PO_4^{3-}$) form [35,49]. When the

pH of the solution was lower than the $pH_{pzc}$ of the adsorbent, the positively charged surface of the adsorbent supported the removal of phosphate ions. This designates the involvement of the electrostatic force of attraction between the anionic species of phosphates ($H_2PO_4^-$, $HPO_4^{2-}$) and the positively charged surface of the adsorbent in phosphate removal under acidic conditions. However, when the solution pH was more significant than the adsorbent's $pH_{pzc}$, the adsorbent's negatively charged surface decreased the removal of phosphate ions. This points towards the contribution of repulsive forces between the negatively charged surface of the adsorbent and phosphate species or competition between $OH^-$ with $PO_4^{3-}$ ions under alkaline conditions, which ultimately results in a decrease in percent removal of phosphate ions onto $TW\text{-}Fe_3O_4$ [35,50].

The most probable adsorption mechanisms which control the phosphate adsorption on $TW\text{-}Fe_3O_4$ surface can be described in reactions (3) and (4). The protonated surface of the adsorbent will stabilize anions such as $H_2PO_4^-$ and $HPO_4^{-2}$ through electrostatic interaction when the solution pH is lower than PZC, which governs the formation of outer-sphere complex between the phosphate and $TW\text{-}Fe_3O_4$ surface.

$$Fe\text{-}OH_2^+ + H_2PO_4^- \leftrightarrow Fe\text{-}OH_2^+ \text{------} H_2PO_4^- \tag{3}$$

$$Fe\text{-}OH_2^+ + HPO_4^{-2} \leftrightarrow Fe\text{-}OH_2^+ \text{------} HPO_4^{-2} \tag{4}$$

Similar reaction mechanisms for phosphate anions adsorption have also been reported by many researchers [49,51].

### 3.2.3. Influence of Adsorbent Dosage

The determination of possible effective dose for adsorption has been ensured, as the removal efficiency depends upon the availability of active sites of adsorbent. The results indicating the effect of adsorbent dosage on the percentage removal of phosphate by TW-$Fe_3O_4$ are shown in Figure 3b. It could be predicted from the figure that the percentage removal of phosphate was increased from 50.4 to 91.2% in parallel to the adsorbent dose ranging from 0.2 to 1.2 g. The maximum percentage removal of 92% was achieved when the amount of absorbent was 1.2 g. The increase in adsorption with increasing adsorbent dosage could be attributed to the surge in available active sites and surface area of adsorbent that provided a significant number of adsorption sites. However, the removal efficiency of phosphate remained unchanged when the adsorbent dose increased from 1.2 to 1.5 g. This could be attributed to the splitting in the flux or concentration gradient of phosphate in the solution phase and phosphate concentration on the surface of the adsorbent, which causes hindrance of mass transfer process [17]. Therefore, 1.2 g of TW-$Fe_3O_4$ adsorbent was optimized for conducting further all adsorption experiments.

### 3.2.4. Effect of Contact Time

The impact of the contact time of the adsorbent with the phosphate ions present in the aqueous solution was investigated, and the results are shown in Figure 4a. The results showed that the percentage removal of phosphate gradually increased from 50 to 91.8% when contact time increased from 5 to 75 min. After 75 min of reaction time, the adsorption process became stable and no further increase in percentage removal was observed. The maximum removal efficiency of 91.8% was achieved in 75 min of contact time. After the establishment of equilibrium, the value of percentage removal became almost constant due to the occupation of active sites by phosphate ions on the surface of the adsorbent and the gradual reduction in the concentration gradient between the bulk solution and adsorbent.

### Application of Kinetic Models

In order to evaluate the kinetic date for phosphate adsorption onto TW-$Fe_3O_4$, four different types of kinetic models, such as pseudo-first-order, pseudo-second-order and intra-particle diffusion models, were used in the forms as mentioned below:

$$\ln(q_e - q_t) = ln q_e - k_1 t \tag{5}$$

$$\frac{t}{q_t} = \frac{1}{k_2 q_{e2}} + \frac{t}{q_t} \tag{6}$$

$$q_t = k_p t^{1/2} + C \tag{7}$$

where $q_e$ and $q_t$ are the quantity of phosphate adsorbed at equilibrium and time t ($mg \cdot g^{-1}$). The $k_1$ is the pseudo-first-order rate constant ($min^{-1}$), $k_2$ is the pseudo-second-order rate constant ($g \cdot mg^{-1} \cdot min^{-1}$), $k_u$ is the film diffusion rate constant ($min^{-1}$) and $k_p$ is the intra-particle diffusion rate constant. The corresponding kinetic parameters and linear plots of the four kinetics models are presented in Table 2 and Figure 4b–d, respectively. The high regression coefficient values ($R^2$) indicated that the phosphate adsorption onto TW-Fe$_3$O$_4$ followed both pseudo-second order and intra-particle diffusion models well. The $q_{e \cdot cal}$ (mg/g) from the pseudo-second-order model (82.64 mg/g) was found to be higher than $q_{e \cdot cal}$ (mg/g) from the pseudo-first-order model (81.23 mg/g). The lower value of $k_2$ ($1.73 \times 10^{-4}$) than $k_1$ ($3.27 \times 10^{-2}$) signifies that the rate of adsorption was very slow. Further, the lower value of $R^2$ for the pseudo-first-order model as compared to the pseudo-second-order model revealed that adsorption is a surface phenomenon. In a previous study, As(III) adsorption on the surface of metal oxide was reported to follow the chemical process [52]. However, in this study, it was indicated that adsorption of phosphate on TW-Fe$_3$O$_4$ adsorbent is controlled by a pseudo-second-order reaction mechanism along with intra-particle diffusion. It is also noticeable from Figure 4d that the linear plot of intra-particle diffusion does not pass through the origin, which reveals that the mechanism of PO$_4{}^{3-}$ adsorption on TW-Fe$_3$O$_4$ is complex, and both the surface adsorption and intra-particle diffusion govern the rate-determining step [53].

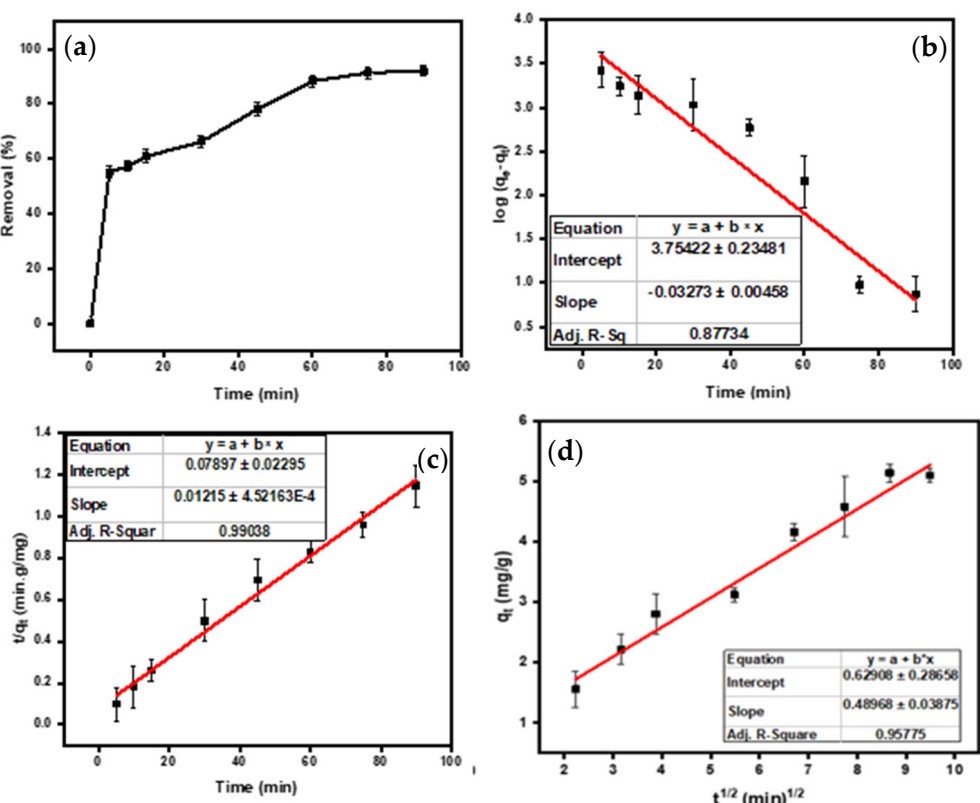

**Figure 4.** (**a**) Effect of contact time; (**b**) pseudo-first-order plot; (**c**) pseudo-second-order plot; (**d**) intra-particle diffusion plot.

**Table 2.** Kinetic parameters for adsorption of $PO_4^{3-}$ ions by TW-$Fe_3O_4$ adsorbent.

| Reaction Order | Parameters | Values |
|---|---|---|
| Pseudo first order | $Ci$ (mg/L) | 5 |
| | $k_1$ (min$^{-1}$) | $3.27 \times 10^{-2}$ |
| | $q_{e.cal}$ (mg/g) | 81.23 |
| | $R^2$ | 0.9197 |
| Pseudo second order | $k_2$ (g·mg$^{-1}$·min$^{-1}$) | $1.73 \times 10^{-4}$ |
| | $q_{e.cal}$ (mg/g) | 82.64 |
| | $R^2$ | 0.9889 |
| Intra-particle Diffusion | $k_p$ (min$^{-1}$) | $1.73 \times 10^{-4}$ |
| | $q_{e.cal}$ (mg/g) | 81.23 |
| | $R^2$ | 0.9197 |

### 3.2.5. Effect of Temperature

The effect of temperature on the adsorption capacity of $PO_4^{3-}$ was studied and the results are shown in Figure S2a. It is apparent from the figure that an increase in solution temperature had a positive effect on the adsorption of $PO_4^{3-}$ ions by TW-$Fe_3O_4$ from 298 to 323 K, thereby confirming the endothermic nature of adsorption in this temperature range. The percentage removal of $PO_4^{3-}$ ions by TW-$Fe_3O_4$ at 298, 303, 313 and 323 K was observed at 68, 74, 85 and 92%, respectively, revealing that an increase in temperature could increase the adsorption potential of TW-$Fe_3O_4$ adsorbent. In contrast, the adsorption potential decreased dramatically with increasing temperature from 323 to 333 K, which could be due to the physical nature of adsorption at the higher reaction temperature. The current findings clearly indicate that the higher temperatures did not favor $PO_4^{3-}$ removal by adsorption on TW-$Fe_3O_4$ adsorbent.

The thermodynamic parameters, including Gibbs free energy change ($\Delta G$), enthalpy change ($\Delta H$) and entropy change ($\Delta S$), were examined, and the calculated values for all thermodynamic parameters are presented in Table 3. The $\Delta H$ and $\Delta S$ values were determined using the well-known Van't Hoff equation:

$$InKc = \frac{\Delta H}{RT} - \frac{\Delta S}{R} \tag{8}$$

$$Kc = \frac{X}{Ce} \tag{9}$$

where Kc is the adsorption equilibrium constant, *T* is the temperature (*K*) and *R* is the general gas constant (8.314 J/K.mol). A linear thermodynamic plot is shown in Figure S2b, where $\Delta H$ and $\Delta S$ values were determined from the slope ($\Delta H/R$) and intercept ($\Delta S/R$) of the line, respectively. The $\Delta G$ (Table 3) was obtained by using the following Gibbs free energy equation:

**Table 3.** Thermodynamic parameters for adsorption of $PO_4^{3-}$ ions by TW-$Fe_3O_4$.

| Temperature (*K*) | $\Delta G$ (kJ/mol) | $\Delta H$ (kJ/mol) | $\Delta S$ (J/Kmol) |
|---|---|---|---|
| 298 | −1.6141 | | |
| 303 | −1.9100 | 4.2664 | 19.733 |
| 313 | −2.1073 | | |
| 323 | −2.3046 | | |

Table 3 shows that the values of $\Delta G$ are negative at all temperatures and decrease with increasing temperature, showing the favorable and spontaneous adsorption process. The positive $\Delta H$ and $\Delta S$ values suggested the endothermic nature of the adsorption process and an increase in randomness at the solid–solution interface during the adsorption process. In the current investigation, the trend for thermodynamic parameters is similar to that previously reported by several researchers using the same magnetized tea waste adsorbent for removal of Cr(VI), U(IV) and As(III) ions [34,37,38,42].

### 3.2.6. Effect of Adsorbate Concentration

The initial concentration of phosphate ions has shown an extreme effect on adsorption properties. The results depicted in Figure 5a revealed that the percentage removal of phosphate on TW-Fe$_3$O$_4$ adsorbent decreased from 92 to 69% when the initial phosphate concentration was increased from 1 to 20 mg/L. This may be because of the ratio between surface active sites of the adsorbent and the concentration of phosphate ions decreased at a higher initial concentration, resulting an overall decrease in the percentage removal efficiency of phosphate on TW-Fe$_3$O$_4$.

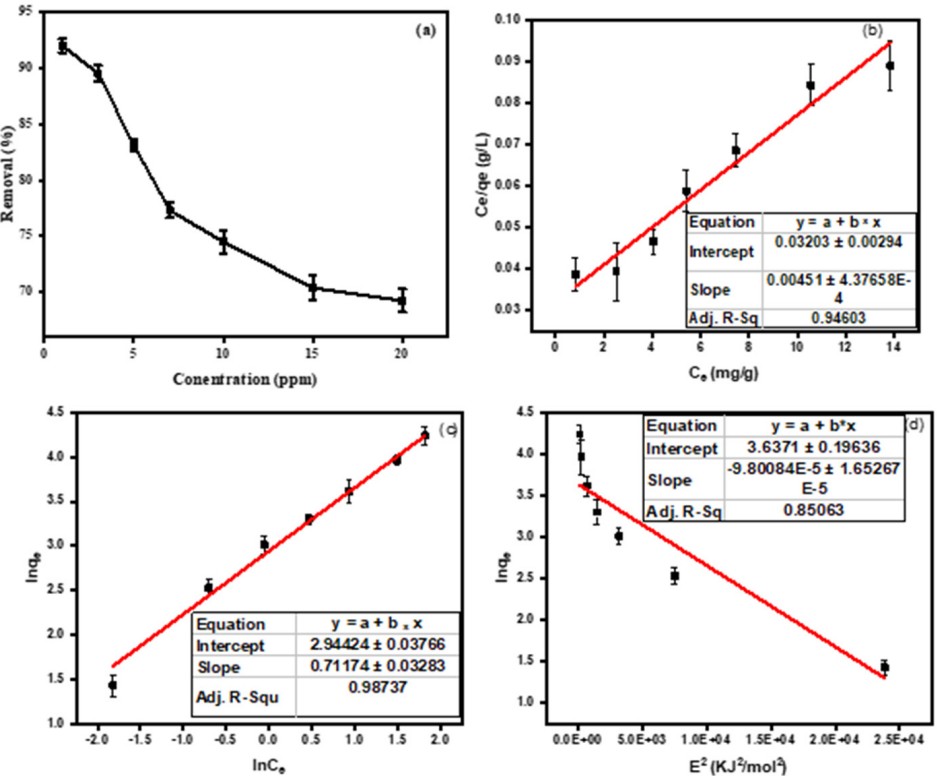

**Figure 5.** (**a**) Effect of phosphate concentration; (**b**) plot of Langmuir model; (**c**) plot of Freundlich model; (**d**) plot of DRK model.

### 3.2.7. Application of Adsorption Isotherm Models

To evaluate the adsorption mechanism of phosphate onto TW-Fe$_3$O$_4$, different adsorption isotherm models such as Langmuir, Freundlich and Dubinin–Radushkevich (DRK) models were employed on the adsorption equilibrium data (Table S1). The Langmuir model assumes monolayer adsorption on a homogenous surface of the adsorbent, while the Freundlich model assumes multilayer adsorption on a heterogeneous surface of the adsorbent. The DRK model presupposes that adsorption occurs through a physical, chemical or simply ion exchange. The equations for the Langmuir, Freundlich and DRK models are shown below in Equations (10)–(12), respectively. The linear form of model for the Langmuir adsorption isotherm is given as:

$$ln q_e = \frac{C_e}{qm} + \frac{1}{kLqm}\, C_e \tag{10}$$

where $C_e$ is the concentration at equilibrium (mg/L), $q_e$ is the quantity of adsorbate adsorbed per unit mass of adsorbent (mg/g), $q_m$ is a maximum adsorption capability of adsorbent (mg/g) and $K_L$ is the binding energy constant showing the interaction between adsorbate and adsorbent. Similarly, the linear form of the Freundlich adsorption isotherm is given as:

$$lnq_e = lnK_F + \frac{1}{n}lnC_e \qquad (11)$$

where $K_F$ is the Freundlich constant showing the adsorption capacity and *1/n* is the heterogeneity factor. The linear form of the DRK adsorption isotherm can be expressed with the following equation:

$$lnq_e = lnq_m - \beta\varepsilon^2 \qquad (12)$$

In Equation (13), $\varepsilon$ is Polanyi adsorption potential and is equal to:

$$\varepsilon = RT\left\{1 + \frac{1}{C_e}\right\} \qquad (13)$$

The value of energy of adsorption determines whether the process is physical or chemical adsorption, which is related to the value of $\beta$ $(mol^2J^{-2})$ as shown by the following relationship:

$$E = \frac{1}{(2\beta)^{\frac{1}{2}}} \qquad (14)$$

According to literature reports, if the energy of adsorption $E$ is below 8 kJ·mol$^{-1}$, between the range of 8 to 16 kJ·mol$^{-1}$ or higher than 16 kJ·mol$^{-1}$, then it is physical adsorption, ion exchange and chemical adsorption, respectively [54]. The plots of different models are shown in Figure 5b–d. It can be revealed from Figure 5c that the Freundlich model was best applicable to the equilibrium data with a higher regression coefficient ($R^2$ = 0.987) value compared to other applied models in this study. This suggested a heterogeneous surface and multilayer adsorption mechanism. Furthermore, the magnitude of the Freundlich constant (*n*) was greater than 1 (1.849), indicating the favorable nature of phosphate adsorption by TW-Fe$_3$O$_4$ adsorbent. Similar results for multilayer adsorption of phosphate on lanthanum-loaded magnetic carbon-shell adsorbent has been reported by other researchers [35]. In the current investigation, the adsorption process was observed to be physiosorption, since the energy of adsorption E (3.53 kJ/mol) was lower than 8 kJ·mol$^{-1}$. This is in accordance with the proposed adsorption mechanism for an outer-sphere complex involving weak electrostatic forces between the phosphate and active sites on the TW-Fe$_3$O$_4$ surface. In the current investigation, the calculated value of $E$ was very close to that reported for As(III) removal using the same magnetized tea waste adsorbent, and the authors also suggested the same physical sorption mechanism [34].

### 3.3. Regeneration of Adsorbent

The reusability of the TW-Fe$_3$O$_4$ adsorbent was assessed by the regeneration of the phosphate-adsorbed TW-Fe$_3$O$_4$ after adsorption. TW-Fe$_3$O$_4$ was first separated by filtration after the adsorption experiment. The functional sites of the TW-Fe$_3$O$_4$ adsorbent were then regenerated by mixing 1.2 g of phosphate-absorbed TW-Fe$_3$O$_4$ with 50 mL of NaOH (1 M) over 2 h. Afterward, the adsorbent was dried in a vacuum oven at 70 °C before reusing it in the batch study. The results of the reusability of TW-Fe$_3$O$_4$ for phosphate adsorption are shown in Figure 6, which showed that the percent removal efficiency of phosphate adsorption onto TW-Fe$_3$O$_4$ decreased gradually from 92 to 85% after six consecutive adsorption–desorption cycles. The decrease in removal efficiency could be due to the loss of functional sites and partial desorption [55]. This result indicated that TW-Fe$_3$O$_4$ can be successfully reusable for phosphate removal from water solutions within a given time. The concentration of Fe leached from TW-Fe$_3$O$_4$ after phosphate adsorption was also detected in the range of 0.079–0.032 mg/L from the first to sixth adsorption–desorption cycles. This confirmed that a minute quantity of iron was entered in leachate, which was lower than the recommendable limit in water (0.3 mg/L) [56]. Thus, it was concluded that during the usage of TW-Fe$_3$O$_4$ adsorbent for phosphate removal, iron will not create any environmental problems and the synthesized adsorbent has recommendable stability and reusability.

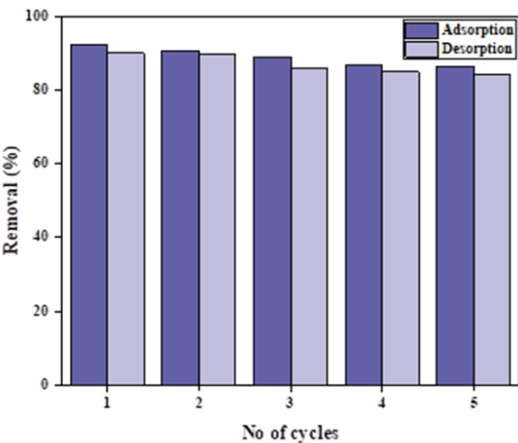

**Figure 6.** % Removal of $PO_4^{3-}$ versus adsorption–desorption cycles.

### 3.4. Comparison of Adsorbent with Other Adsorbents

To evaluate the potentially effective adsorbent for phosphate adsorption, the maximum adsorption capacity of our adsorbent TW-Fe$_3$O$_4$ in the present investigation was compared with various types of modified adsorbents available in the literature (Table 4). The data tabulated in Table 4 show that the adsorption capacity of TW-Fe$_3$O$_4$ for phosphate removal was much higher than those of other modified adsorbents. This confirmed that TW-Fe$_3$O$_4$ is a promising and economical adsorbent for the removal of phosphate from aqueous solutions.

**Table 4.** Comparison with other adsorbents used for adsorption of $PO_4^{3-}$ ions from aqueous solutions.

| Adsorbents | Adsorption Capacity $q_e$ (mg/g) | References |
|---|---|---|
| Fe-loaded litchi Chinese seed waste | 100 | [57] |
| Orange waste gel loaded with zirconium | 57 | [58] |
| Zirconia-loaded lignocellulosic butanol residue | 7.17 | [59] |
| Iron-impregnated coir pith | 70.90 | [60] |
| Giant reed-based adsorbent | 54.67 | [61] |
| Amine-crosslinked wheat stalk | 60.61 | [62] |
| Amine-crosslinked cotton stalk | 51.15 | [62] |
| Iron oxide tailing | 8.21 | [63] |
| Iron hydroxide eggshell | 14.4 | [64] |
| Blast furnace slag | 8.00 | [65] |
| Lanthanum-treated lignocellulosic sorbent | 20.04 | [66] |
| Activated fly ash | 58.92 | [67] |
| Magnetic-oxide-impregnated tea waste (TW-Fe$_3$O$_4$) | 226.76 | Present Study |

### 4. Conclusions

The tea waste was successfully modified by Fe$_3$O$_4$ to enhance adsorption capabilities towards phosphate anions. The adsorbents were successfully characterized through a variety of techniques. The structural modification of tea waste was confirmed by XRD, where two diffractions were detected which were the characteristics of magnetite. The adsorption of phosphate was performed at pH 5. The removal study indicated that the adsorption capacity of phosphate anions on TW-Fe$_3$O$_4$ was enhanced when the adsorbent dosage and reaction time were enhanced. The Langmuir adsorption capacity for phosphate ions by TW-Fe$_3$O$_4$ was determined to be 226.76 mg/g. The Freundlich adsorption isotherm model was suitable for describing the adsorption of phosphate onto TW-Fe$_3$O$_4$. The pseudo-second-order and intra-particle diffusion models showed a good fit for the adsorption kinetics of phosphate ions on TW-Fe$_3$O$_4$ with high regression coefficient values

of 0.987 and 0.989, respectively. The good fitting of adsorption data to the adsorption kinetics and isotherm models also confirmed that the removal of phosphate by TW-Fe$_3$O$_4$ occurred through physical adsorption. The current investigation proved that TW-Fe$_3$O$_4$ is an inexpensive, environmentally friendly and readily available adsorbent with a high adsorption capacity for phosphate removal and could be successfully employed for six adsorption–desorption cycles.

**Supplementary Materials:** The following supporting information can be downloaded at: https://www.mdpi.com/article/10.3390/w15203541/s1, Figure S1: Plot for PZC for TW-Fe$_3$O$_4$. Figure S2: Effect of temperature and Van't Hoff Plot for removal of PO$_4^{3-}$ ions by TW-Fe$_3$O$_4$. Table S1: Equilibrium parameters for adsorption of PO$_4^{3-}$ ions by TW-Fe$_3$O$_4$ adsorbent.

**Author Contributions:** Conceptualization, K.H.S. and M.W.; methodology, K.H.S., M.F. (Misbah Fareed), M.W. and S.S. (Shabnam Shahida); formal analysis, M.F. (Muhammad Fahad), S.S. (Sadaf Sarfraz), N.S.S. and M.W.; investigation, M.R.H., A.B. and T.A.; resources, N.S.S. and M.R.H.; data curation, K.H.S. and S.S. (Shabnam Shahida) writing—original draft preparation, K.H.S., M.F. (Misbah Fareed), S.S. (Sadaf Sarfraz) and A.B.; writing—review and editing, N.S.S., K.H., M.R.H., C.H. and M.W.; visualization, N.S.S., C.H. and M.W., supervision, K.H.S. and S.S. (Sadaf Sarfraz); funding acquisition, K.H.S., M.R.H. and C.H. All authors have read and agreed to the published version of the manuscript.

**Funding:** The authors received funding from the HEC, Pakistan through NRPU (Project No. 8817), King Saud University Riyadh, Saudi Arabia through the Researchers Supporting Project number (RSP2023R222) and the Korean government (MSIT) through National Research Foundation of Korea (NRF) (No. 2021R1A2C1093183 and 2021R1A4A1032746).

**Data Availability Statement:** All the data for this manuscript is provided in the manuscript in Figures and Tables form.

**Acknowledgments:** Higher Education of Pakistan (HEC) supported the current research work through the National Research Program for Universities (NRPU), Project No. 8817. The authors acknowledge HEC for successfully providing funds to complete this research work. The authors are also grateful to the Researchers Supporting Project number (RSP2023R222), King Saud University, Riyadh, Saudi Arabia, for financial support. CH acknowledges the support of the National Research Foundation of Korea (NRF) grant funded by the Korean government (MSIT) (No. 2021R1A2C1093183 and 2021R1A4A1032746).

**Conflicts of Interest:** The authors declare no conflict of interest.

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
