# Peer review of "Tea-Waste-Mediated Magnetic Oxide Nanoparticles as a Potential Low-Cost Adsorbent for Phosphate (PO43−) Anion Remediation"

_water, doi:10.3390/w15203541_

Round 1
Reviewer 1 Report
1. The authors are invited to express the most significant and innovative acts of this article to support the publication of the article in the journal.
2. The introduction does not present the scientific problem well, which makes the article not very readable and scientific. it is recommended to reorganize the language and scientific problem, and to introduce the background and the scientific problem of the study in a concise and rigorous scientific language.
3. The article lists a large amount of data, but most of the data are not core data and do not reflect the scientific problems and innovative results. it is suggested to put some non-core data into the supporting information to make the article more concise.
4. The conclusion section does not summarize the results of the work done in the article well, and the overly complicated language makes it impossible for readers to quickly grasp the information they want to obtain.
5. The author's language needs to be refined, and the language in the whole article is not standardized enough.
6. Suggested citations Chin. J. Chem. 2023, 41, 672, Aggregate 2022, 3, e308, which have a discussion of environmental aspects that can be drawn on.
The language of the article should be more concise, clear, and scientific.
Author Response
"Please see the attachment."

Reviewer 2 Report
This study focuses on the use of magnetic oxide nanoparticles impregnated in tea waste (TW-Fe3O4) as an adsorbent for the removal of phosphate ions (PO43-) from aqueous solutions. The research involves a comprehensive characterization of the TW-Fe3O4 nano-adsorbent using various analytical techniques, including FE-SEM, TEM, EDX, BET, FTIR, and XRD. Overall, this study presents the development and characterization of a novel nano-adsorbent, TW-Fe3O4, which exhibits exceptional efficiency in removing phosphate ions from aqueous solutions. The research provides valuable insights into the optimization of adsorption conditions and the adsorption mechanisms, highlighting the potential of this nano-adsorbent for addressing phosphate pollution in water systems.
I think this paper could be published after minor revisions on this journal.
Here are some minor comments.
1. Please change title of y axis to charge or define what is “delta pH
2. Please provide reference XRD peak for Fe3O4 in Figure 2b.
3. Please provide chemical manufacture or dealer in part 2.1
4. Please provide more details in Part 2.2 (e.g., the amount of materials used)
5. Add the removal data at 0 min in Figure 5a.
It is acceptable.
Author Response
"Please see the attachment."

Reviewer 3 Report
1. I am not happy with the title. Thus, the title must convert to: "Waste-tea mediated Magnetic oxide nanoparticles as a potential low-cost adsorbent for phosphate (PO4 -3) anions remediation"
2. The English and grammatical mistakes should be revised.
3. The novelty of this study should be highlighted in the last paragraph of introduction.
4. Introduction needs to be little bit more elaboration about the need of green nanotechnology. Therefore, importance of green nanotechnology and their diverse applications should be mentioned in a paragraph. More literature study needed with recent references in the Introduction.
5. The authors must explain in a separate section what are the superiorities of Fe3O4 NPs over the other NPs? For this purpose, cite the following article;
Environmental Nanotechnology, Monitoring & Management, 17, p.100661.
6. The difference between green synthesis methods and chemical methods should be added to the manuscript. Also, the author/authors should mention the green methods types and why plant extract method is better? The author must use the following articles:
a. PLoS One, 17(8), p.e0268184.
b. IEEE Transactions on NanoBioscience, 22(2), pp.308-317.
7. What is the significance of waste-tea extract in synthesize of Fe3O4 nanoparticles?
8. The mechanism of formation Fe3O4 NPs using phytochemicals in green synthesis must be explained. The author should use the following articles:
a. SN Applied Sciences, 2(5), p.991.
b. Journal of Materials Science: Materials in Electronics, 31, pp.11303-11316.
c. Current Organic Synthesis, 17(7), pp.558-566.
d. Micro & Nano Letters 15, no. 6 (2020): 415-420.
9. The authors should add recyclability and reusability of the Fe3O4 NPs.
10. The comparison of the other similar research with this work in a table.
11. The author/authors must explain the role of waste-tea extract in controlling the morphology of the prepared Fe3O4 NPs.
12. Add the Ref. for the Preparation of waste-tea Extract.
13. What is the role of pH, temperature, plant type, concentration and other parameters in the green synthesized method?
14. More explanation about XRD spectra of TW and TW-Fe3O4 is required.
15. The graphical preparation (Real photograph from the lab.) of Fe3O4 NPs should be added to the manuscript with all steps.
16. How the author/authors detect the main phytochemicals in waste-tea extract?
17. Fig.3 shows that the author did not able to synthesize nanosized NPs and this is explaining why I need to see the real photo of the products.
18. Fig. 2 also a bit odd for me and I guess it follows the same issues that have been mentioned before.
19. The Conclusions section should include:
a. A highlight of your hypothesis, new concepts and innovations.
b. A summary of key improvements compared to findings in literature
c. Your vision for future work
20. The list of Ref. must be updated. The following articles must be cited in the revised manuscript:
- International Nano Letters, 12(2), pp.153-168.
- Desalination Water Treat 190, 179-192
- Journal of Environmental Chemical Engineering 11, no. 2 (2023): 109514.
- Nanoscience & Nanotechnology-Asia 12 (6), 27-38
- Jordan Journal of Physics 13, no. 2 (2020): 123-135.
- Emergent Materials 5 (3), 683-693
- International Nano Letters, 12(2), pp.139-151.
21. Please replace the out-of-date Refs with the above-mentioned Ref.
Regards and good luck.
1. I am not happy with the title. Thus, the title must convert to: "Waste-tea mediated Magnetic oxide nanoparticles as a potential low-cost adsorbent for phosphate (PO4 -3) anions remediation"
2. The English and grammatical mistakes should be revised.
3. The novelty of this study should be highlighted in the last paragraph of introduction.
4. Introduction needs to be little bit more elaboration about the need of green nanotechnology. Therefore, importance of green nanotechnology and their diverse applications should be mentioned in a paragraph. More literature study needed with recent references in the Introduction.
5. The authors must explain in a separate section what are the superiorities of Fe3O4 NPs over the other NPs? For this purpose, cite the following article;
Environmental Nanotechnology, Monitoring & Management, 17, p.100661.
6. The difference between green synthesis methods and chemical methods should be added to the manuscript. Also, the author/authors should mention the green methods types and why plant extract method is better? The author must use the following articles:
a. PLoS One, 17(8), p.e0268184.
b. IEEE Transactions on NanoBioscience, 22(2), pp.308-317.
7. What is the significance of waste-tea extract in synthesize of Fe3O4 nanoparticles?
8. The mechanism of formation Fe3O4 NPs using phytochemicals in green synthesis must be explained. The author should use the following articles:
a. SN Applied Sciences, 2(5), p.991.
b. Journal of Materials Science: Materials in Electronics, 31, pp.11303-11316.
c. Current Organic Synthesis, 17(7), pp.558-566.
d. Micro & Nano Letters 15, no. 6 (2020): 415-420.
9. The authors should add recyclability and reusability of the Fe3O4 NPs.
10. The comparison of the other similar research with this work in a table.
11. The author/authors must explain the role of waste-tea extract in controlling the morphology of the prepared Fe3O4 NPs.
12. Add the Ref. for the Preparation of waste-tea Extract.
13. What is the role of pH, temperature, plant type, concentration and other parameters in the green synthesized method?
14. More explanation about XRD spectra of TW and TW-Fe3O4 is required.
15. The graphical preparation (Real photograph from the lab.) of Fe3O4 NPs should be added to the manuscript with all steps.
16. How the author/authors detect the main phytochemicals in waste-tea extract?
17. Fig.3 shows that the author did not able to synthesize nanosized NPs and this is explaining why I need to see the real photo of the products.
18. Fig. 2 also a bit odd for me and I guess it follows the same issues that have been mentioned before.
19. The Conclusions section should include:
a. A highlight of your hypothesis, new concepts and innovations.
b. A summary of key improvements compared to findings in literature
c. Your vision for future work
20. The list of Ref. must be updated. The following articles must be cited in the revised manuscript:
- International Nano Letters, 12(2), pp.153-168.
- Desalination Water Treat 190, 179-192
- Journal of Environmental Chemical Engineering 11, no. 2 (2023): 109514.
- Nanoscience & Nanotechnology-Asia 12 (6), 27-38
- Jordan Journal of Physics 13, no. 2 (2020): 123-135.
- Emergent Materials 5 (3), 683-693
- International Nano Letters, 12(2), pp.139-151.
21. Please replace the out-of-date Refs with the above-mentioned Ref.
Regards and good luck.
Author Response
"Please see the attachment."

Round 2
Reviewer 1 Report
Suggested acceptance
Suggested acceptance
Reviewer 2 Report
previous comments have been addressed.
Reviewer 3 Report
The authors revised the manuscript according to our comments. Therefore, the manuscript can be accepted in this current version.